# Efficient inter-species conjugative transfer of a CRISPR nuclease for targeted bacterial killing

Thomas A. Hamilton[1], Gregory M. Pellegrino [1], Jasmine A. Therrien[1], Dalton T. Ham[1], Peter C. Bartlett [1], Bogumil J. Karas[1], Gregory B. Gloor [1*] & David R. Edgell[1*]

The selective regulation of bacteria in complex microbial populations is key to controlling pathogenic bacteria. CRISPR nucleases can be programmed to kill bacteria, but require an efficient and broad-host range delivery system to be effective. Here, using an *Escherichia coli* and *Salmonella enterica* co-culture system, we show that plasmids based on the IncP RK2 conjugative system can be used as delivery vectors for a TevSpCas9 dual nuclease. Notably, a *cis*-acting plasmid that encodes the conjugation and CRISPR machinery conjugates from *E. coli* to *S. enterica* with high frequency compared to a *trans* system that separates conjugation and CRISPR machinery. In culture conditions that enhance cell-to-cell contact, conjugation rates approach 100% with the *cis*-acting plasmid. Targeting of single or multiplexed sgRNAs to non-essential genes results in high *S. enterica* killing efficiencies. Our data highlight the potential of *cis*-acting conjugative plasmids as a delivery system for CRISPR nucleases or other microbial-altering agents for targeted bacterial killing.

---

[1] Department of Biochemistry, Schulich School of Medicine and Dentistry, London, ON N6A5C1, Canada. *email: ggloor@uwo.ca; dedgell@uwo.ca

Microbial ecosystems are essential for human health and proper development, and disturbances of the ecosystem correlate with a multitude of diseases[1–8]. A central problem is the lack of specific tools to selectively control pathogenic species, or to otherwise alter the composition of the human microbiome and other microbial communities. Traditional methods such as antibiotic treatment suffer from a number of limitations that preclude selective control in a defined and efficient manner, and are becoming less effective because of overuse and the development of multidrug resistant bacteria[9]. Phage-based therapy is limited by host range and the rapid development of phage-resistant bacteria[10]. Probiotics and prebiotics are effective but of use in only a few defined conditions[11]. Stool transplants are effective treatments for gastrointestinal dysbioses, but can result in widespread alterations in the composition of the gut microbiome with unknown long-term effects[12–14]. These limitations highlight an increasing need for effective and selective tools for the targeted modulation of microbiomes.

CRISPR (clustered regularly interspaced short palindromic repeats) is a bacterial immune system that targets invading DNA for elimination[15–18]. The Cas9 protein (CRISPR-associated protein 9) has been adapted for genome-editing applications in a wide range of organisms[19]. Cas9 and related proteins can also be used as antimicrobial agents because the sequence of the guide RNA can be changed to target Cas9 to specific sequences in bacterial genomes. The introduction of double-strand breaks in bacterial chromosomes by Cas9 causes replication fork collapse and subsequent cell death[20–22]. A critical component of studies adapting CRISPR as a sequence-specific antimicrobial was the testing of different delivery vectors, including well-studied conjugation systems that would mobilize CRISPR-containing plasmids. However, the low frequency of conjugation was found to be a limiting factor in CRISPR-mediated killing, whereas phagemid- or bacteriophage-mediated delivery was found to be much more efficient. Nonetheless, conjugative plasmid delivery of CRISPR nucleases remains an attractive option because conjugative plasmids have broad-host ranges[23], are resistant to restriction-modification systems[24], are easy to engineer with large coding capacities[25], and do not require a cellular receptor[26] that would provide a facile mechanism for bacterial resistance. Conjugative plasmids are known to encode factors that promote biofilm formation[27] presumably because enhanced cell-to-cell contact increases rates of conjugative plasmid transfer[28]. Conjugative plasmids may thus be well suited for delivery of molecular tools for modulating composition of human microbial communities[29–32], many of which exist as biofilms.

Here, we show that conjugative plasmids are an efficient system to deliver CRISPR nucleases to bacteria. We develop a cis-conjugative system where the plasmid encodes both the conjugative machinery and CRISPR nuclease[33], as opposed to previously tested trans setups where the conjugative machinery and nuclease were encoded on separate DNA molecules[20] (Fig. 1). Bacteria that receive the cis-conjugative plasmid become potential donors for subsequent rounds of conjugation, potentially leading to exponentially increasing numbers of conjugative donor bacteria in the population. We test the cis-conjugative plasmid in a two-species co-culture system, finding high frequency of conjugative transfer of plasmids from Escherichia coli to Salmonella enterica under conditions that enhance cell-to-cell contact. Our results highlight the promise of conjugative delivery of CRISPR nucleases as an effective tool for modification of microbiomes.

## Results

**Increased conjugation frequency with a cis-conjugative plasmid.** We constructed a conjugative plasmid, pNuc, based on the IncP RK2[34] plasmid to examine parameters that contributed to conjugation (Fig. 1a). The pNuc plasmid encoded the TevSpCas9 nuclease (I-TevI nuclease domain fused to Streptococcus pyogenes Cas9[33]) controlled by an arabinose-inducible pBAD promoter[35], and a single-guide RNA (sgRNA) cassette driven by a constitutive promoter derived from the tetracycline resistance gene (pTet) into which we cloned oligonucleotides corresponding to predicted target sites in the S. enterica genome (Fig. 1b, Supplementary Data 1). Two forms of the plasmid were constructed (Fig. 1a, Supplementary Fig. 1, Supplementary Datas 5–7). First, a cis configuration (pNuc-cis) where the origin of transfer (oriT) and CRISPR system were cloned into the pTA-Mob backbone that encodes the genes necessary for conjugation[34]. The second setup employed a plasmid trans configuration (pNuc-trans) that included only the CRISPR system, oriT, and chloramphenicol resistance. The oriT sequence on pNuc-trans is recognized by the relaxase expressed in trans from the pTA-Mob helper plasmid to facilitate conjugation. The pNuc-trans setup mimics the plasmids used in previous studies that examined conjugative delivery of CRISPR nucleases in an E. coli donor/recipient system[20–22].

We used the pNuc-cis and pNuc-trans plasmids to test the hypothesis that the cis setup would support higher levels of conjugation relative to the trans setup in a time-course filter-mating assay using E. coli as the donor and S. enterica as the recipient (Fig. 1c). As shown in Fig. 1d, conjugation frequency (transconjugants/total recipients) for pNuc-cis continually increased over the time of the experiment reaching a maximum of $1 \times 10^{-2}$ by 24 h. In contrast, conjugation frequency for pNuc-trans peaked at early time points with a maximal frequency of $\sim 1 \times 10^{-3}$, declining to $\sim 1 \times 10^{-5}$ by 24 h. We isolated five S. enterica transconjugants each from experiments with the pNuc-cis or pNuc-trans plasmids and showed that the transconjugants were viable donors for subsequent conjugation of the pNuc-cis plasmid to naive recipients, but not for the pNuc-trans plasmid (Fig. 1e). Furthermore, higher frequency conjugation of pNuc-cis was not due to higher copy number relative to pNuc-trans in the E. coli donor or S. enterica transconjugants (Fig. 1f), or because pNuc-cis was significantly more stable than pNuc-trans (Fig. 1g).

To determine if longer incubation times resulted in higher conjugation frequency with the pNuc-cis system, we used a liquid conjugation assay consisting of low-salt LB (LSLB) media into which varying ratios of donor E. coli and recipient S. enterica cells were added. After 72 h incubation at 37 °C with mild agitation at 60 RPM, we found that high donor to recipient ratios (1:1, 10:1, and 50:1) yielded more transconjugants per recipient than experiments with lower donor to recipient ratios (1:5 or 1:10) (Supplementary Fig. 2a). We also showed that decreasing the NaCl concentration of the media to 0.25% w/v resulted in an increased conjugation frequency at a 10:1 donor:recipient ratio (Supplementary Fig. 2b). Using the 10:1 donor:recipient ratio, and 0.25% NaCl LSLB media, we examined the effect of culture agitation on conjugation, finding that both 0 and 60 RPM resulted in similar conjugation frequencies while a higher 120 RPM resulted in lower conjugation frequency (Supplementary Fig. 2c).

Collectively, these data show that pNuc-cis has an $\sim 1000$-fold higher conjugation frequency than the pNuc-trans system at 24 h post-mixing because bacteria that receive pNuc-cis become donors for subsequent rounds of conjugation. This would lead to exponentially increasing numbers of conjugative donors in the population. Thus, our data differ significantly from previous studies that concluded that conjugation frequency with a trans system was a limiting factor for CRISPR delivery[20].

**Cell-to-cell contact significantly increases conjugation.** The previous experiments demonstrated that pNuc-cis was more efficient at conjugation in a filter mating assay on solid media. To

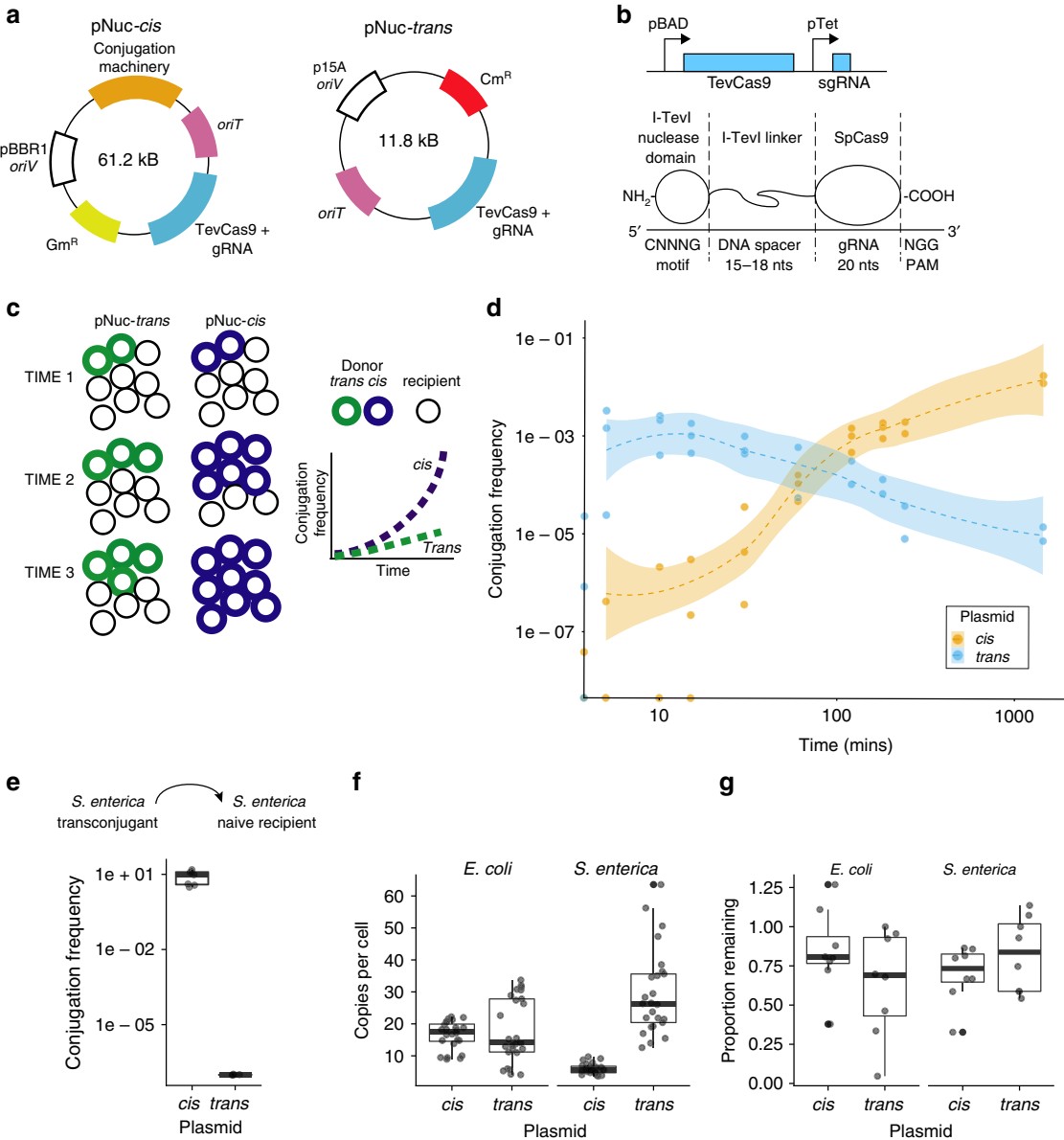

**Fig. 1** Impact of *cis* or *trans* localization of conjugative machinery on conjugation frequency. **a** Schematic view of the pNuc-*cis* and pNuc-*trans* plasmids. *oriT* conjugative origin of transfer, *oriV* vegetative plasmid origin, Gm^R gentamicin resistance gene, Cm^R chloramphenicol resistance gene, TevSpCas9/sgRNA coding region for TevSpCas9 nuclease gene and sgRNA. Conjugative machinery, genes required for conjugation derived from the IncP RK2 conjugative system. **b** (Top) The TevSpCas9 and sgRNA cassette (not to scale) highlighting the arabinose regulated pBAD and constitutive pTet promoters. (Below) The modular TevSpCas9 protein and DNA binding site. Interactions of the functional TevSpCas9 domains with the corresponding region of substrate are indicated. **c** Model of pNuc spread after conjugation with the *cis* and *trans* setups. Cell growth overtime will account for increase of pNuc-*trans*. **d** Filter mating assays performed over 24 h demonstrate that pNuc-*cis* has a higher conjugation frequency than pNuc-*trans*. Points represent independent experimental replicates, and the 95% confidence intervals are indicated as the shaded areas. Conjugation frequency is reported as the number of transconjugants (Gm^R, Kan^R) per total recipient *S. enterica* cells (Kan^R). **e** Conjugation frequency of *S. enterica* transconjugants harboring either pNuc-*cis* or pNuc-*trans* to naive *S. enterica* recipients. Data are shown as boxplots with points representing individual replicate experiments. **f** pNuc-*cis* and pNuc-*trans* copy number determined by quantitative PCR in either *E. coli* or *S. enterica*. Data are shown as boxplots with solid lines indicating the median of the data, the rectangle the interquartile bounds, and the wiskers the range of the data. Points are individual experiments. **g** pNuc-*cis* and pNuc-*trans* stability in *E. coli* or *S. enterica* determined as the ratio of cells harboring the plasmid after 24 h growth without antibiotic selection over total cells. Data are shown as boxplots with dots indicating independent experiments. Source data are provided as a Source Data file

test whether liquid culture conditions that enhanced cell-to-cell contact through biofilm formation resulted in increased conjugation with pNuc-*cis*, we included 0.5 mm glass beads in liquid cultures that would provide a solid surface for cell-to-cell contact[36–38] and observed conjugation frequencies as high as 100% with pNuc-*cis* (Fig. 2a, b). This conjugation frequency represents a ~500- to 1000-fold enhancement compared to the solution or filter-based pNuc-*cis* assays. Increasing culture agitation to 60 RPM had no discernible effects on conjugation frequency with pNuc-*cis*. With the pNuc-*trans* plasmid, conjugation frequency

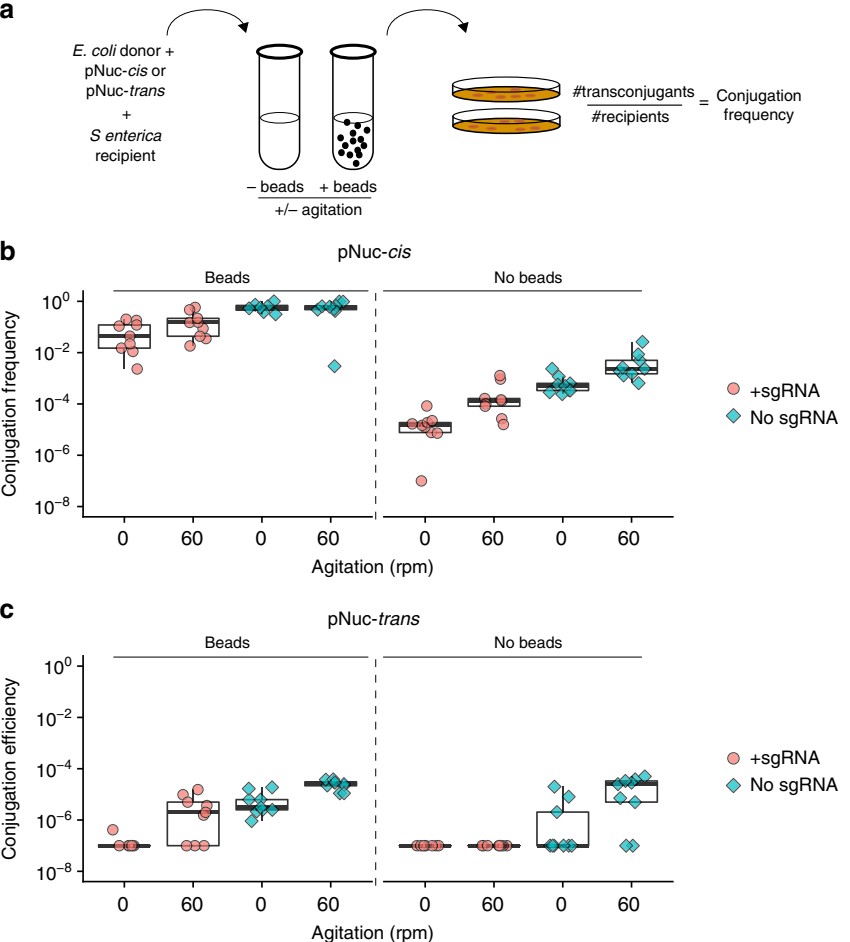

**Fig. 2** Influence of enhanced cell-to-cell contact on conjugation frequency. **a** Schematic of experimental design. Liquid conjugation experiments in culture tubes with **b** pNuc-*cis* and **c** pNuc-*trans* were performed with 0.5 mm glass beads or without glass beads (filled diamonds) over 72 h at the indicated shaking speed (in revolutions per minute). Conjugations were performed with (filled circles) or without (filled diamonds) sgRNA targeting the STM1005 locus cloned into pNuc-*cis* and pNuc-*trans*. Both plasmids encoded the TevSpCas9 nuclease. Data are plotted on a log10 as boxplots with data points from independent biological replicates. The solid line represents the median of data, the rectangle represents the interquartile range of the data, and the whiskers represent the maximum and minimum of the data. Source data are provided as a Source Data file

ranged from $1 \times 10^{-8}$ to $1 \times 10^{-4}$ (Fig. 2b), supporting the hypothesis that gains in conjugation frequency with the pNuc-*cis* system resulted from exponentially increasing number of cells that become donors for subsequent rounds of conjugation after receiving the plasmid.

Interestingly, we observed a reduction in conjugation frequency when a *S. enterica* specific sgRNA was cloned onto pNuc-*cis* (the + guide condition) (Fig. 2a). We postulate that a proportion of *S. enterica* are killed immediately post-conjugation. We attribute this killing to leaky expression of the TevSpCas9 nuclease from the pBAD promoter under repressive culture conditions (+0.2% glucose).

**S. enterica killing by conjugative delivery of Cas9 and sgRNAsS.** To demonstrate that the TevSpCas9 nuclease could be delivered by conjugation to eliminate specific bacterial species, we designed 65 total sgRNAs targeting 38 essential genes, 23 non-essential genes, and 4 genes with unresolved phenotypes (Fig. 3a, Supplementary Data 1). The 65 sgRNA sites were arrayed around the *S. enterica* chromosome (Fig. 3b), differed in their relative position within each gene, and what strand was being targeted. We assessed the efficacy of each sgRNA in killing *S. enterica* by comparing the ratio of *S. enterica* colony counts under conditions

where TevSpCas9 expression from the pBAD promoter was induced with arabinose or repressed with glucose. Using *E. coli* as the conjugative donor, we found a range of *S. enterica* killing efficiencies between 1 and 100% (Fig. 3a). To demonstrate that the I-TevI nuclease domain could function in the context of other Cas9 orthologs, we fused the I-TevI nuclease domain to SaCas9 from *Staphylococcus aureus* to create TevSaCas9. SaCas9 differs from SpCas9 in possessing a longer PAM requirement[39]. With TevSaCas9 we observed high killing efficiency (93 ± 8%, mean ± standard error) when TevSaCas9 was targeted to the *fepB* gene of *S. enterica* (Supplementary Fig. 3). sgRNAs expressed as pairs from separate promoters also yielded high killing efficiencies (Supplementary Fig. 4), demonstrating the potential for multiplexing guides to overcome mutational inactivation of individual guides. Sampling *S. enterica* colonies resistant to killing from experiments with different sgRNAs revealed three types of escape mutants: nucleotide polymorphisms in the chromosome target site that would weaken sgRNA–DNA interactions, transposable element insertions that inactivated sgRNA expression, and rearrangements of pNuc that impacted TevSpCas9 function (Supplementary Fig. 5)[40].

We considered a number of variables that would influence sgRNA killing efficiency in *S. enterica*, including predicted sgRNA activity according to an optimized prokaryotic model[41], targeting

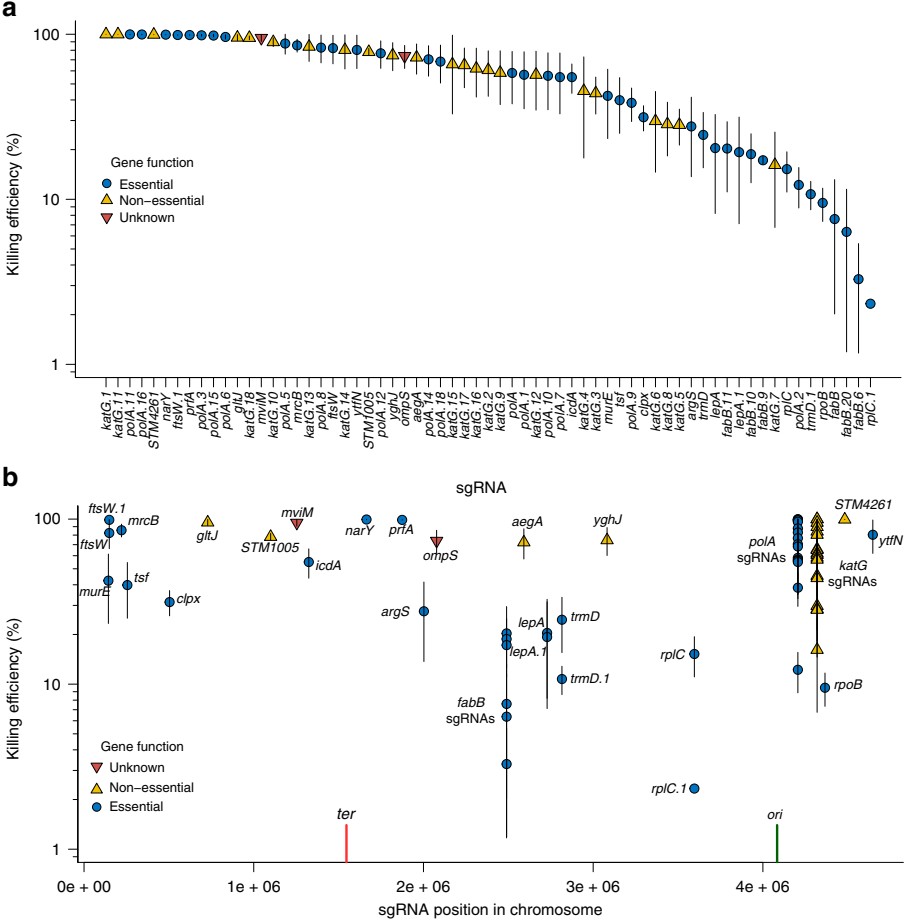

**Fig. 3** Killing efficiency of sgRNAs targeted to the *S. enterica* genome. **a** Ranked killing efficiency of individual sgRNAs coded as to whether the target site in found in an essential gene (blue filled circles), nonessential gene (orange diamonds), or unknown if the gene is essential (inverted red triangles). Vertical lines represent the standard error of the data from at least three biological replicates. **b** Killing efficiency of each sgRNA plotted relative to their position in the *S. enterica* genomes, color-coded as in panel **a**. The terminator region (*ter*) and origin of replication (*ori*) are indicated by vertical red and green lines, respectively. Source data are provided as a Source Data file

of the sense or anti-sense strands for transcription, the relative position of the sgRNA in the targeted gene, targeting of the leading or lagging replicative strands, and the essentiality of the targeted gene. Taken independently, no single variable was strongly correlated with sgRNA killing efficiency (Fig. 4 and Supplementary Fig. 6). A generalized linear model was used to assess the significance of each variable on sgRNA killing efficiency, revealing that sgRNA score positively correlated with predicted activity ($p < 0.02$, $t$ test) while targeting essential genes was negatively correlated with killing efficiency ($p < 0.03$, $t$ test) (Supplementary Fig. 6). The moderate statistical support from the linear model suggests that a robust understanding of parameters that influence sgRNA targeting and activity in prokaryotic genomes remains a work in progress, particularly in the context of conjugative plasmids.

During the course of these experiments, we noted that some sgRNAs were recalcitrant to cloning (Supplementary Fig. 7). In particular, sgRNAs targeting essential genes in *S. enterica* were more likely to yield inactive clones than sgRNAs targeting nonessential genes (Supplementary Data 2). Whole plasmid sequencing revealed no insertions in 15 clones with sgRNAs targeting nonessential genes, whereas 7/13 clones sgRNAs targeting essential genes had insertions. These findings suggest that leaky expression of the TevSpCas9 nuclease from the pBAD promoter is sufficient to cause cellular toxicity in *E. coli*, and selection for inactive plasmids. Thus, choosing sgRNAs with

minimal identity and off-target sites in the *E. coli* genome (Supplementary Data 3) will facilitate conjugative delivery of sgRNAs and CRISPR nucleases.

## Discussion

A central problem in microbiology and infectious disease control is the lack of tools to alter the composition of microbial communities or to control pathogenic species. One crucial concept in microbiome manipulation is that complete elimination of the target organism(s) is not required to restore the community because the constituent organisms of a bacterial population exhibit exponential growth[42]. It is only necessary to reduce the relative abundance of the target organism below a threshold to achieve control. CRISPR-based nucleases can be easily repurposed as sequence-specific antimicrobial agents, yet the development of a robust and broadly applicable delivery system remains a key milestone.

In this study, we adapted an IncP RK2 conjugative plasmid to deliver specific functional sequences to species of interest. Previous studies recognized the potential of conjugative delivery of CRISPR nucleases, emphasizing improvements in frequency as key to future applications[20]. Our study differs from previous attempts in one key facet—we used a *cis* setup where the pNuc plasmid encoded the conjugative machinery as well as the TevSpCas9 nuclease. The pNuc-*cis* plasmid promotes increased

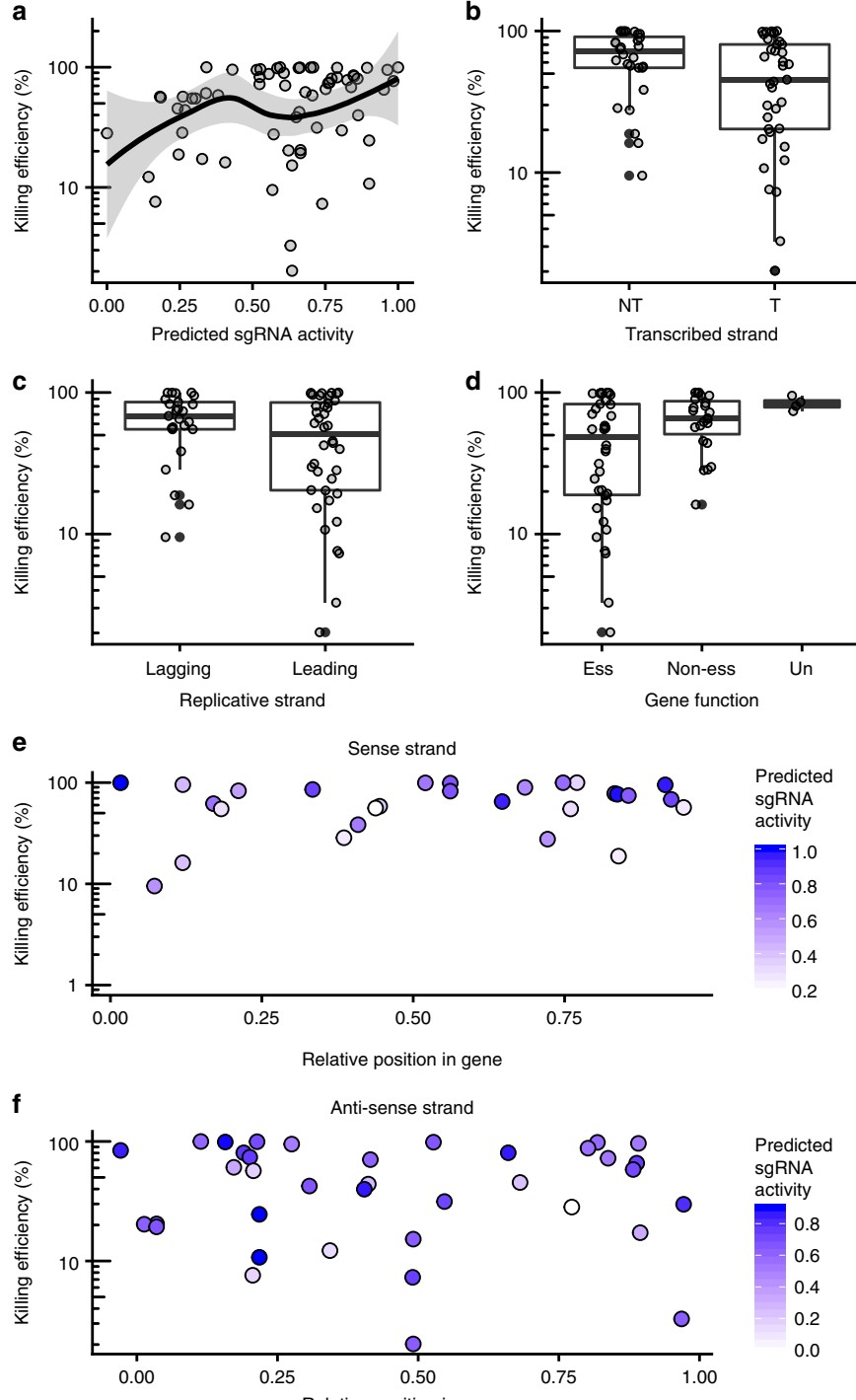

**Fig. 4** Effect of sgRNA targeting parameters on killing efficiency. **a** Plot of predicted sgRNA activity versus *S. enterica* killing efficiency for all 65 sgRNAs. The shaded area is the 95% confidence interval of the line of best fit. Boxplots of sgRNAs targeting different strands for **b** transcriptional (S sense strand, AS anti-sense strand) and **c** replication, and **d** sgRNAs targeting genes with essential (Ess), nonessential (NEss), or unresolved phenotypes (Un) versus killing efficiency. **e** Plot of relative position of sgRNAs within genes versus average killing efficiency for the sense strand and **f** anti-sense strand of targeted genes. For each plot, points are filled according to their predicted sgRNA activity. Killing efficiency is plotted on a log10 scale. Source data are provided as a Source Data file

occurrence of conjugation events because our data infers that transconjugants become donors for subsequent reconjugation, leading to significant increases in conjugation frequency relative to the pNuc-*trans* plasmid. Previous studies employed strains with the conjugative machinery embedded in the chromosome of the donor bacteria (similar to the pNuc-*trans* setup), meaning that only a single round of conjugation could occur. In our two-species *E. coli*–*S. enterica* system, we observed conjugation frequencies approaching ~100% with pNuc-*cis* in culture conditions that promoted cell-to-cell contact and biofilm formation. Because the IncP RK2 system can be conjugated to a wide diversity of bacteria[43], and because conjugative systems are widespread in

bacteria, our system in theory could be used to deliver the TevSpCas9 nuclease (or other CRISPR nuclease) in complex microbial communities.

It is possible that conjugation may not be the limiting factor in all systems. Indeed, improving regulation of TevSpCas9 to prevent cellular toxicity will improve conjugation efficiencies and counter negative selection on pNuc for inactivating mutations. Our data suggest that parameters that govern sgRNA activity in bacterial systems are poorly understood. Other factors, including compatibility with resident plasmids[44], expression of CRISPR and conjugation genes in diverse bacteria, and targeting of conjugative plasmids by naturally occurring CRISPR systems[45], may also be relevant. Many of these issues have defined molecular solutions, such as broad-host range plasmid origins, redundant sgRNAs, universal promoters, and codon optimization for gene expression. Anti-CRISPR proteins[46,47] that are specific for relevant CRISPR systems could also be included on pNuc-*cis* to prevent or reduce acquisition of CRISPR-mediated resistance. We also envision using multiple strains of donor bacteria harboring versions of pNuc-*cis* based on different conjugative plasmid backbones[48], each encoding redundant programmable CRISPR nucleases or other microbial-modulating agents or sequences.

Microbial communities have complex bacterial compositions and they inhabit diverse environments. Many human microbial communities exist as biofilms[1], which presents a challenge for delivery of antimicrobial agents. Indeed, a number of disease conditions result from microbial imbalances in mucosal surfaces that are dominated by biofilms, for example *Clostridium dificile* infection[49]. Rates of conjugation can be high in biofilms[28] and conjugative plasmids express factors that promote biofilm formation to enhance cell-to-cell contact necessary for formation of the conjugative pilus[27]. By using a donor bacteria that is a native resident of the target biofilm the pNuc-*cis* plasmid could be introduced to microbial communities more readily than delivery vectors that have difficulty penetrating biofilms. Conversely, other delivery vectors, such as phage-based methods, are better suited to planktonic conditions where conjugation is less efficient. Depending on the nature of the microbiome and dysbiosis, a combination of conjugative- and phage-based CRISPR delivery systems may be appropriate.

## Methods

**Bacterial and yeast strains.** *E. coli* EPI300 (F′ λ⁻ *mcrA* Δ(*mrr-hsdRMS-mcrBC*) φ80d*lacZ*ΔM15 Δ(*lac*)X74 *recA1 endA1 araD139* Δ(*ara*, *leu*)7697 *galU galK rpsL* (Str^R) *nupG trfA dhfr*) (Epicenter) was used for cloning and as a conjugative donor. *Salmonella typhimurium* sub. species *enterica* LT2 (Δ*hilA*::Kan^R) (acquired from Dr. David Haniford at Western University) was used as a conjugative recipient strain. *S. cerevisiae* VL6-48 cells (*MAT***a**, *his3*Δ*200*, *trp*Δ*1*, *ura3-52*, *ade2-101*, *lys2*, *psi* + *cir*°) was used for yeast assembly of conjugative plasmids.

**Plasmid construction.** Plasmids were constructed using a modified yeast assembly[50,51]. A list of primers is provided in Supplementary Data 4. The pNuc-*trans* plasmid was constructed by polymerase chain reaction (PCR) amplification of fragments with 60–120 bp homology overlaps from pre-existing plasmids. The *oriT* fragment was amplified from pPtGE30[52] using primers DE-3302 and DE-3303. The p15A origin, chloramphenicol acetyl-transferase gene, and sgRNA cassette was amplified using primers DE-3308 and DE-3309 from a modified pX458 plasmid containing the TevSpCas9 coding region[33]. The TevSpCas9 gene was amplified from the modified pX458 plasmid using primers DE-3306 and DE-3307. The *araC* gene and pBAD promoter were amplified from pBAD-24[35] using primers DE-3304 and DE-3305. The CEN6-ARSH4-HIS3 yeast element was amplified from pPtGE30[52] using primers DE-3316 and DE-3317. *S. cerevisiae* VL6-48 was grown from a single colony to an OD_600 of 2.5–3, centrifuged at 2500×*g* for 10 min and washed in 50 mL sterile ddH₂0 and centrifuged. Cells were resuspended in 50 mL of 1 M sorbitol, centrifuged, and spheroplasting initiated by resuspending the pellet in 20 mL SPE solution (1 M sorbitol, 10 mM sodium phosphate buffer pH 7, 10 mM Na₂EDTA pH 7.5) and by adding 30 μL 12 M 2-mercaptoethanol and 40 μL zymolyase 20T solution (200 mg zymolyase 20T (USB), 9 mL H₂O, 1 mL 1 M Tris pH 7.5, 10 mL 50% glycerol) and incubated at 30 °C with shaking at 75 RPM. The yeast was considered spheroplasted once the ratio of the OD_600 in sorbitol to the OD_600 of yeast in ddH₂0 reached 1.8–2. Spheroplasts were centrifuged at 1000×*g*

for 5 min before being gently resuspended in 50 mL 1 M sorbitol, and centrifuged again. Spheroplasts were then resuspended in 2 mL STC solution (1 M sorbitol, 10 mM Tris-HCl pH 7, 10 mM CaCl₂) and incubated at room temperature for 10 min. Pooled DNA fragments at equimolar ratio for each plasmid assembly were gently mixed with 200 μL of spheroplasted yeast and incubated at room temperature for 10 min. A volume of 1 mL of PEG-8000/CaCl₂ solution (20% (w/v) PEG 8000, 10 mM CaCl₂, 10 mM Tris-HCl, pH 7.5) was added and incubated at room temperature for 20 min before being centrifuged at 1500×*g* for 7 min. Yeast was resuspended in 1 mL of SOS solution (1 M sorbitol, 6.5 mM CaCl₂, 0.25% (w/v) yeast extract, 0.5% (w/v) peptone) and incubated at 30 °C for 30 min. The spheroplast solution was added to 8 mL of histidine-deficient regenerative agar (Teknova), poured into a petri dish, and incubated overnight at 30 °C. A volume of 8 mL histidine-deficient liquid regenerative media was then added on top of the solidified regenerative agar and grown at 30 °C for 2–5 days. Total DNA was isolated from 1.5 to 3 mL *S. cerevisiae* using 250 μL buffer P1 (50 mM Tris-HCl pH 8.0, 10 mM EDTA, 100 μg/mL RNase A), 12.5 μL zymolyase 20 T solution and 0.25 μL 12 M 2-mercaptoethanol and incubated at 37 °C for 1 h. In total, 250 μL buffer P2 (200 mM NaOH, 1% sodium dodecyl sulfate) was added, incubated at room temperature for 10 min, followed by addition of 250 μL buffer P3 (3.0 M CH₃CO₂K pH 5.5). DNA was precipitated with 700 μL ice-cold isopropanol, washed with 70% ethanol, briefly dried and resuspended in 50 μL sddH₂O. The plasmid pool was subsequently electroporated into *E. coli* EPI300. Individual colonies were screened by diagnostic digest (Supplementary Fig. 7) and sequencing (Supplementary Data 2), and one clone for each sgRNA selected for further use. TevSpCas9 sgRNAs targeting *S. enterica* genes were predicted as previously described[33]. A TevSpCas9 site consists of (in the 5′ to 3′ direction) an I-TevI cleavage motif (5′-CNNNG-3′), a DNA spacer region of 14–19 bp separating the I-TevI cleavage site and the SpCas9 sgRNA binding site, and a SpCas9 PAM site (5′-NGG-3′). Putative sites in the *S. enterica* LT2 genome were ranked according to the predicted activity of the identified I-TevI cleavage site (relative to the I-TevI cognate 5′-CAACG-3′ cleavage site) and the fit of the DNA spacer region to nucleotide tolerances of I-TevI. Oligonucleotides corresponding to the guide RNA were cloned into a BsaI cassette site present in pNuc-*trans*. To construct the pNuc-*cis* plasmid, the *oriT*, *araC*, TevCas9, sgRNA, and CEN6-ARSH4-HIS3 elements were amplified from pNuc-*trans* using primers DE-3024 and DE-3025 that possessed 60 bp homology to both sides of the AvrII restriction site in pTA-Mob. The pTA-Mob plasmid was linearized by AvrII (New England Biolabs), combined with the PCR amplified fragment from pNuc-*trans* and transformed into *S. cerevisiae* VL6-48 spheroplasts. Correct pNuc-*cis* clones were identified as above for pNuc-*trans*. Both pNuc-*trans* and pNuc-*cis* were completely sequenced to confirm assembly. A detailed plasmid map and sequence of each plasmid is provided as Supplementary Fig. 1 and Supplementary Data 1–3.

**Quantitative PCR.** *E. coli* EPI300 donors and *S. enterica* transconjugants harboring pNuc-*trans* and pTA-Mob (*trans* helper plasmid) or pNuc-*cis* were grown overnight under selection. sgRNAs were absent from the *cis* and *trans* plasmids. Overnight cultures were diluted 1:50 in selective media and grown to an A_600 of ~0.5. Each culture was diluted, plated on selective LSLB plates (10 g/L tryptone, 5 g/L yeast extract, and 5 g/L sodium chloride, 1% agar), and grown overnight. Colonies were counted manually to determine the CFUs/mL of each culture. At the same time, 500 μL of each culture was pelleted and resuspended in 500 μL 1× phosphate-buffered saline (PBS) and incubated at 95 °C for 10 min before immediate transfer to −20 °C. Quantitative real-time PCR was performed on boil-lysed samples using SYBR Select Master Mix (Applied Biosystems) using primers DE-4635 and DE-4636 that amplified a DNA fragment present on both pNuc-*trans* and pNuc-*cis*. Purified pNuc-*trans* was used as a copy number standard.

**Filter mating conjugation.** Saturated cultures of donor *E. coli* EPI300 and recipient *S. enterica* LT2 were diluted 1:50 into 50 mL nonselective LSLB media. The diluted cultures were grown to an A_600 of ~0.5 and concentrated 100-fold by centrifugation at 4000×*g* for 10 min. A volume of 200 μL of concentrated donors were mixed with 200 μL concentrated recipients on polycarbonate filters adhered to conjugation plates (LSLB supplemented with 1.5% agar). Conjugation proceeded at 37 °C from 5 min to 24 h. Following conjugation, filters were placed in conical tubes containing 30 mL of 1× PBS (8 g/L NaCl. 0.2 g/L KCl, 1.42 g/L Na₂HPO₄, 0.24 g/L KH₂PO₄) and vortexed for 1 min to remove the bacteria from the filter. The supernatant was serially diluted and plated on LSLB plates with selection for donor *E. coli* EPI300 (gentamicin 40 μg/mL for the *cis* setup and gentamicin 40 μg/mL, chloramphenicol 25 μg/mL for the *trans* setup), recipient *S. enterica* LT2 (kanamycin 50 μg/mL), and transconjugants (kanamycin 50 μg/mL, chloramphenicol 25 μg/mL, 0.2% D-glucose for for pNuc-*trans* transconjugants or kanamycin 50 μg/mL, gentamicin 40 μg/mL, 0.2% D-glucose for pNuc-*cis* transconjugants). D-glucoserepresses the expression of TevCas9 in transconjugants. Plates were incubated overnight at 37 °C for 16–20 h. Colonies were counted manually.

**S. enterica to S. enterica conjugation.** *S. enterica* LT2 transconjugants harboring pNuc-*cis* or pNuc-*trans* with no sgRNA encoded were obtained from plate conjugation experiments described in detail in the supplementary methods.

Transconjugant colonies were grown overnight in LSLB supplemented with kanamycin 50 µg/mL, gentamicin 40 µg/mL and 0.2% D-glucose for pNuc-*cis*, or kanamycin 50 µg/mL, chloramphenicol 25 µg/mL and 0.2% D-glucose for pNuc-*trans*. *S. enterica* LT2 was transformed with pUC19 to confer ampicillin resistance for use as a recipient and was grown overnight in LSLB supplemented with kanamycin 50 µg/mL and ampicillin 100 µg/mL. All donor and recipient *S. enterica* cultures were diluted 1:50 into LSLB and grown to an $A_{600}$ of 0.5 before spreading 200 µL of each on a conjugation plate supplemented with 0.2% w/v D-glucose to repress TevSpCas9 expression. Conjugations proceeded for 2 h at 37 °C before cells were scraped into 500 µL SOC with a cell spreader. Resulting cell suspensions were serially diluted and plated to select for donors (kanamycin 50 µg/mL, gentamicin 25 µg/mL for pNuc-*cis* or kanamycin 50 µg/mL, chloramphenicol 25 µg/mL for pNuc-*trans*), recipient (kanamycin 50 µg/mL, ampicillin 100 µg/mL), and transconjugant (kanamycin 50 µg/mL, gentamicin 40 µg/mL, ampicillin 100 µg/mL for pNuc-*cis*, chloramphenicol 25 µg/mL, ampicillin 100 µg/mL for pNuc-*trans*). Plates were incubated at 37 °C for 16–20 h and colonies were counted manually.

**Liquid and bead-supplemented conjugation assays**. *E. coli* EPI300 and recipient *S. enterica* LT2 were grown overnight to saturation. Tubes containing 5 mL LSLB supplemented with 0.2% D-glucose were inoculated with 180 µL saturated *E. coli* and 18 µL saturated *S. enterica*. Bead-supplemented conjugations were prepared similarly with the addition of 1 mL soda lime glass beads (0.5 mm diameter). Conjugations proceeded by incubating at 37 °C with 0 or 60 RPM agitation for 72 h. Cultures were homogenized by vortexing, serially diluted and spot-plated in 10 µL spots on plates containing appropriate antibiotic selection for donors, recipients, and transconjugants. Plates were incubated at 37 °C for 16–20 h. Colonies were counted manually. Alterations to this protocol were made to determine the effect of donor to recipient ratio (50:1, 10:1, 1:1, 1:10, 1:50), NaCl concentration (2.5, 5, and 10 g/L) and shaking speed (0, 60, and 120 RPM) on conjugation frequency.

**Killing efficiency assays**. Saturated cultures of *E. coli* EPI300 donors habouring pNuc-*trans* plasmids encoding sgRNAs and recipient *S. enterica* LT2 were diluted 1:50 into LSLB supplemented with 0.2% D-glucose. The diluted cultures were grown to an $A_{600}$ of ~0.5. 200 µL of each donor was mixed with 200 µL of recipient on a conjugation plate supplemented with 0.2% D-glucose to repress expression of TevSpCas9. Conjugations proceeded for 1 h at 37 °C before cells were scraped into 500 µL SOC (20 g/L tryptone, 5 g/L yeast extract, 0.5 g/L NaCl, 2.5 mM KCl, 10 mM $MgCl_2$, and 20 mM D-glucose) with a cell spreader. Resulting cell suspensions were serially diluted and plated on selection for donors and recipients in addition to selection for transconjugants with CRISPR repression (kanamycin 50 µg/mL, chloramphenicol 25 µg/mL, 0.2% D-glucose) and transconjugants with CRISPR activation (kanamycin 50 µg/mL, chloramphenicol 25 µg/mL, 0.2% L-arabinose). Plates were incubated overnight at 37 °C for 16–20 h. Killing efficiency is the ratio of cells on selective to nonselective plates.

**Escape mutant analyses**. Escape mutant colonies were picked from plates selecting for exconjugant *S. enterica* cells with TevSpCas9 activated after conjugation. These colonies were grown overnight to saturation and plasmids were extracted using the BioBasic miniprep kit. The isolated plasmids were then electroporated into *E. coli* EPI300 cells and re-isolated for analysis. The plasmids were analyzed by diagnostic restriction digest with FspI and MsiI, and by multiplex PCR for the chloramphenicol resistance marker, and a TevSpCas9 gene fragment. Total DNA was isolated using a standard alkaline lysis protocol followed by isopropanol precipitation of the DNA. Potential target sites were PCR amplified from the total DNA sample using Ampliaq 360 (Thermofisher Scientific) and subsequently sequenced.

**sgRNA off-target predictions in *E. coli***. To predict sgRNA off-target sites, we searched the *E. coli* genome for sites with less than six mismatches to each sgRNA using a Perl script with an XOR bit search (provided as Supplementary Software 1). A mismatch score was calculated that indicates the likelihood of a stable sgRNA/DNA heteroduplex using the formula

$$mm\_score = \sum_{mismatch} 0.5^{non\_seed} + 1.2^{seed},$$

where non_seed is a mismatch in the nonseed region of the sgRNA (positions 1–12 from the 5′ end of the target site) and seed is a mismatch in the seed regions (positions 13–20 from the 5′ end of the target site). By this method, mismatches in the 5′ end of sgRNA/DNA heteroduplex are more tolerated than mismatches closer to the PAM sequence. For each sgRNA, we also added a correction for if the adjacent three nucleotides matched the consensus SpCas9 PAM sequence 5′-NGG-3′. Off-target sites with perfect match PAMs were given more weight than off-target sites with 1 or 2 mismatches. Sample fasta formatted files of sgRNAs (sgRNA.test.fa) and an *E. coli* genome (MG16552.fna) are also provided (Supplementary Datas 8 and 9). Source code and instructions to execute the perl script are provided in Supplementary Software 1. A sample output is shown in Supplementary Fig. 8 and the full table of mismatch scores for each sgRNA is found in Supplementary Data 4.

**Modeling *S. enterica* killing efficiency**. To model sgRNA parameters that were predictive of *S. enterica* killing efficiency, we used a generalized linear model in the *R* statistical language with the formula

$$sgRNA_{KE} \sim sgRNA_{score} + sgRNA_{target\,strand} + sgRNA_{repstrand} + sgRNA_{gene\,func} + sgRNA_{reldist},$$

where $sgRNA_{KE}$ is the average killing efficiency for a given sgRNA, $sgRNA_{score}$ is the predicted sgRNA activity score using the algorithm of Guo et al. [41], $sgRNA_{targetstrand}$ is the transcription strand targeted by the sgRNA (sense or antisense), $sgRNA_{repstrand}$ is whether the sgRNA targets the leading or lagging strand, $sgRNA_{genefunc}$ is whether the sgRNA targets an essential or non-essential gene in *S. enterica*, and $sgRNA_{reldist}$ is the position of the sgRNA relative to the AUG codon of the targeted gene. A summary table and graphical output of the model parameters is shown in Supplementary Fig. 6.

**Reporting summary**. Further information on research design is available in the Nature Research Reporting Summary linked to this article.

## Data availability

All data generated and analyzed during this study are included in the published article or provided in the Supplementary Information. The source data underlying Figs. 1a, 2a–d, 6d, h, and 7c and Supplementary Figs. 1a and 5d are provided as a Source Data file.

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

## Acknowledgements
Supported by a CIHR Project Grant (PJT-159708) to D.R.E., G.B.G., and B.J.K., a Microbiome Initiative Grant from the Weston Foundation to G.B.G., D.R.E., and B.J.K., and an Ontario Genomics SPARK Grant to D.R.E. and G.B.G. We thank Benjamin Joris for help with analyzing plasmid sequencing data.

## Author contributions
T.A.H., G.M.P., B.J.K., G.B.G., and D.R.E. conceived the experiments, T.A.H., G.M.P., J.A.T., D.T.H., and P.B., conducted the experiments, T.A.H., G.M.P., P.B., B.J.K., G.B.G., and D.R.E. analyzed the results, T.A.H., G.B.G., and D.R.E. wrote the paper. All authors reviewed the paper.

## Competing interests
The authors declare no competing interests.
