## [Peer Review File · Nature Communications]

Reviewers' comments:

Reviewer #1 (Remarks to the Author):

The manuscript entitled: "Efficient inter-species conjugative transfer of a 1 CRISPR nuclease for targeted bacterial biofilm modulation" optimized conjugative delivery of CRISPR nucleases for targeted modulation of microbiomes using an *Escherichia coli* and *Salmonella enterica* co-culture system. They found that a cis-acting plasmid that encodes the conjugation and CRISPR machinery promotes efficient conjugation, which is 1000-fold more efficient than the pNuc-trans system. They then observed conjugation frequencies as high as 100% when a solid surface for biofilm formation was provided. In the second part of the paper, they tried to optimize the killing efficiency by minimizing guide RNA identity to the conjugative donor genome, using multiplexing guide RNAs, but they did not find correlation of killing efficiency with predicted sgRNA activity, or position within a coding region. Findings of this manuscript add useful information in targeted bacterial biofilm modulation with CRISPR-associated nucleases. In its present form, however, it has serious shortcomings that need to be corrected before the manuscript is accepted for publication. The following major issues need to be addressed first.

1. Statistics data suggests that multiplexed guides had higher killing efficiencies as a group than their single guide constituents, but there was an exception, rplC, with very low killing efficiency even use multiple guides. Is that because of the essentiality of this gene in *E. coli*? How about change the guide sequence identity with *E. coli*?
2. The experiments study the influences of sgRNA identity to the conjugative donor genome on *S. enterica* killing efficiency are very preliminary. The authors gave very few examples on this experiment. Comprehensive design a pool of sgRNAs with different identity to *E. coli*, and test their influences on killing efficiency would be helpful.
3. There have been reported sgRNA predictions model for prokaryotes (ref. 32 as cited by the authors), will that be helpful for the correlation of killing efficiency with predicted sgRNA activity? A predictable killing efficiency is always welcomed.
4. Minor point: Fig.1C was not cited in the paper. And I do not see it is informative in the paper.

Reviewer #2 (Remarks to the Author):

The work reported in this manuscript describes a successful approach to improve the spread of the established CRISPR/Cas9 system, that serves as an anti-microbial agent, in a two-bacteria setup. The placement of the conjugation machinery and the CRISPR system on a conjugative plasmid leads to improved conjugation frequencies hence the improved spread of CRISPR/Cas9 system. The study is worth to publish, however, there are multiple shortcomings. Hence I conclude as 'major revision'.

Below I list both my major and minor comments:

Major comments:

1. I do not see any compelling evidence to justify the use of biofilm statement in the title. To me this study simply reports how the constructed pNuc-cis improves the spread of the CRISPR nuclease, via conjugation, as compared to the pNuc-trans. The use of CRISPR/Cas9 as a DNA-sequence-specific antimicrobial agent has already been reported hence the authors do not need to re-claim this point.

The authors also do not provide :

- . any background to the biofilm aspect, except mentioning a condition 'mimicking biofilm' at the end of introduction. I encourage the authors to provide all the necessary background in the introduction text instead of taking too many shortcuts by citing other publications;
- . any empirical finding to support that the use of beads leading to increased biofilm formation,

except assuming that the increased surface will lead to biofilm formation. I can follow the logic, however, to have such a claim, use of biofilm in the title, experimental biofilm formation has to be provided.

2. The study uses T_{ev}Sp/Sa/Cas9 system however they do not provide any detail on the mode of action of the killing. The authors should provide all the necessary background to help the readers.

3. The authors use inconsistent terminology throughout the manuscript that leads to confusion. For instance, due to the inconsistent terminology use, it is not clear to me what the authors aim to do with the developed system: Is the aim controlling, killing, modifying, modulating bacteria? All these can be achieved, however, this being a research article, and not a commentary, a better focus is necessary.

Another terminology: conjugation efficiency vs. conjugation frequency. The authors use both terminology interchangeably without a clear distinction. The authors do not perform any genetic engineering to the tra clusters/genes hence there is no 'direct attempt' to improve the efficiency of conjugation. Testing different agitation speed and NaCl concentrations may justify the use of term 'conjugation efficiency', however, due to inconsistent term use I cannot distinguish it. What has been reported is the improved conjugation frequency, or the number of successful conjugation events (given what the conjugation frequency calculation is based on).

4. The methods section is incomplete hence not satisfactory. There is a lack of focus to the important details (see my comments below to specific examples.)

I cannot figure out, neither based on the text in the main body nor in the method section, what has been done for the construction of pNuc-cis. From the Figure 1a, it seems that the authors cloned the CRISPR system and the oriT sequence to pTA-Mob, however, there is a discrepancy between the description in the main body text and the method section. Overall, the method section should clearly describe the specific design of the study and provide clear and concise description of the procedures that were performed.

5. Given the number of plasmids constructed, it would be helpful to provide the nucleotide sequences of the plasmids (or plasmid maps). I would encourage the authors to provide the sequences in supplementary, as gene bank files, or deposit them to open public repositories for easy access.

Minor comments:

L12. Microbial ecosystems are essential for human health and proper development, and disturbances of the ecosystem correlate with a multitude of diseases^{1–5}. → This is a confusing sentence: the citation one is about rhizosphere (I do not see the immediate relevance of rhizosphere on human development), and two to five are about gut microbiomes. Would not it be easier by simply referring to gut microbiome instead of microbial ecosystems?

L20-21 – composition of the microbial ecosystem... → composition of gut microbiome...

Inconsistent terminology for the set goal:

L22, ...targeted modification of microbiomes;

L34, ...targeted modulation of microbiomes;

L42, ...tool for manipulation of microbiomes.

L28-30 – Highly efficient. → Please re-phrase the sentence. It is not clear what is efficient; is it the killing-effect or the conjugation (which rather should be frequency)?

L34 – Here, we optimized conjugative delivery of CRISPR nucleases for targeted modulation of microbiomes. → I do not understand the concept of modulation of microbiomes. This study only shows the use of CRISPR system as an antimicrobial agent. Hence the only control is the killing

effect that has been reported for a single strain. Please re-phrase.

L36 – conjugation machinery <-> conjugative machinery. Please stick to one.

L36-37 – the conjugation machinery and nuclease were separated. → It is not clear what separated means? Are they located on separate plasmids, or a certain part is on the host genome, or ...?

L38 – re-conjugation → conjugation.

L40 – highly efficient conjugative transfer (of ...?) → The object is missing.

L41 – ...mimics a biofilm. → There is no background provided to the biofilm aspect in the introduction?

L41 – ...highlight the promise of... → please simplify.

Figure 1 legend, the first line – I am not familiar with the concept of using statements as figure names. Regardless, the statement is not clear. Efficient in what sense? Would not it be better to give the figure a name that reflects both the content and the message it aims to convey?

Figure 1 legend – a. Schematic of the... → a. Schematic view of the.

Figure 1 legend – ...conjugation genes → There are no genes that are called conjugation genes hence the use 'transfer functions', 'genes required for conjugation' or alike would suit better.

Figure 1 legend – from the pTA-Mob plasmid and derived from the IncP RK2 conjugative system. – It is not clear, what is derived from what.

Figure 1 legend – the b section in the legend is not describing all the elements that are depicted in the b part of the figure.

Figure 1 legend – the section c is erroneous. Firstly, the legend does not describe the figure. Secondly, why does the number of donor cells (green circles) under the use of pNuc-trans is increasing over time? In the -trans setup the pTA-Mob is not mobile, hence to me the number of donors should not increase over time hence this part of the figure is confusing.

Figure 1 – Conjugation efficiency → Please see my comment below on the inconsistent term use.

L44 – ...higher levels of conjugation → This is not clear as there is no highness or lowness to conjugation. A suggestion would be: A cis conjugative plasmid leads to higher conjugation frequency than a plasmid with trans setup.

L45 – We engineered a conjugative plasmid, pNuc, based on the IncP RK2 conjugative plasmid to examine parameters that contributed to conjugation (Fig. 1A). → Please re-phrase. It is not clear what has been engineered. Do the authors refer to pNuc-cis? pNuc-cis looks as if it is based on pTA-Mob, but I do not see which part of pNuc-trans originates from pTA-Mob.

L45 – IncP RK2 conjugative plasmid → Please re-phrase. The citation refer to the pTA-Mob plasmid, which is not conjugative as it does not harbour an oriT.

L46 – The pNuc plasmid encoded... → Please re-phrase, the plasmid contains/carries/harbours the nuclease, does not encodes it.

L48-50 – and a single guide RNA (sgRNA) cassette driven by a constitutive promoter derived from the tetracycline resistance gene (pTet) into which we cloned oligonucleotides corresponding to predicted target sites in the *S. enterica* genome. → Too long sentence, please break it up. Also it

reads as if the oligos were cloned into the tet gene.

L51-54 – This is a lengthy and confusing sentence. Unless I missed it, the method section does not provide the details, the authors simply cloned both the CRISPR system and oriT into the pTA-Mob. Hence, if this is correct, I do not understand why the conjugation machinery and the gentamicin resistance gene were cloned into the pTA-Mob?

L52; L56 – the conjugative assistance plasmid, pTA-Mob; pTA-Mob mobilization plasmid → Please be consistent with the terminology use. I would opt for the term 'conjugative assistance/helper plasmid'.

L61 – conjugation efficiency → This refers to the conjugation frequency and not efficiency. This applies to the entire text.

L64 – This result suggests pNuc-trans is lost from a percentage of exconjugants after the initial burst of conjugation, possibly because it is toxic or unstable. → Why do the authors claim that pNuc-trans was lost from the recipients? I cannot see any experimental findings showing the actual number of donors that are Cm resistant (carrying the pNuc-trans plasmid) over the course of the conjugation experiment. In the absence of such data this is not justifiable.

L75 – conjugation efficiency → conjugation frequencies...

L76 – conjugation efficiency → conjugation frequency.

Figure 2 legend – circles → circles

Figure 2 – I imagine the TevSpCas9 was used to generate the data, but there is nothing mentioned specifying which of the Cas9 system was used to generate the data used in figure 2.

L82 – increase in conjugation... → increase in conjugation frequency...

L82 – cell-cell contact → cell-to-cell contact

L83 – The previous experiments demonstrated that pNuc-cis was more efficient (in compared to what) in a filter mating assay on solid media.

L105 – TevSa/SpCas9 → There is no background provided to neither of the Cas9 systems. What are their differences and why have they been used? Please help the readers to follow your experimental logic.

L110 – A Mann-Whitney test showed → The details of this statistics analysis has not been provided in the methods section.

L118 – growth phenotype → There is no detail provided on how the growth was measured, under what condition(s). There is also no corresponding section in the materials and methods part.

L139 – the gene → the coding sequence? Please be specific wherever possible as 'gene' is a rather vague term.

L140 – predicted sgRNA activity → I do not understand the term 'activity' here. Different sgRNA simply target different positions within the coding sequence of the gene. There is no detail provided on the output of the PERL script hence I cannot understand what is meant by activity. Please specify.

L157 – IncP RK2 conjugative plasmid → Please see my L45 comment above.

L160 – key facet - we used → key facet'em-dash'we used

L175 – broad-host range ↔ broad host range, please stick to one.

L179, L187 – pNUC → pNuc

L241 – I assume that the *S. enterica* has Kan^R phenotype? No background information has been provided on this point. Please help the readers.

L242 – exconjugant cells under TevSpCas9 repressive conditions (kanamycin 50 µg/mL, chloramphenicol 25 µg/mL, and 0.2% glucose) → This part of the methodology only applies to pNuc-trans, since pNuc-cis does not harbour a Cm^R gene? If yes, then the details of the pNuc-cis is missing.

L243; 245 – exconjugant cells expressing TevSpCas9; killing efficiency experiments → I do not understand what the difference is? These are two different protocols for the same experiment? When the arabinose is present in the medium won't they become the same?

L270 – plasmid expression for analysis and re-isolated for analysis. → Please re-phrase.

Supplementary Figures

196493_0_supp_3534160_pm1z48.pdf → Update the title used in the document.

Also the file names are not 'human understandable'. I would prefer file names such as Supplementary Figure 1 and so on. However I do not know who to blame, whether the submission system of the journal or the authors. Regardless, this makes the review even more challenging.

To conclude, the entire text needs a substantial editing prior to re-submission.

Rahmi Lale

Below, we address each reviewer's comments point-by-point. Our responses are in red text.

Reviewer 1.

The manuscript entitled: Efficient inter-species conjugative transfer of a 1 CRISPR nuclease for targeted bacterial biofilm modulation optimized conjugative delivery of CRISPR nucleases for targeted modulation of microbiomes using an Escherichia coli and Salmonella enterica co-culture system. They found that a cis-acting plasmid that encodes the conjugation and CRISPR machinery promotes efficient conjugation, which is 1000-fold more efficient than the pNuc-trans system. They then observed conjugation frequencies as high as 100% when a solid surface for biofilm formation was provided. In the second part of the paper, they tried to optimize the killing efficiency by minimizing guide RNA identity to the conjugative donor genome, using multiplexing guide RNAs, but they did not find correlation of killing efficiency with predicted sgRNA activity, or position within a coding region. Findings of this manuscript add useful information in targeted bacterial biofilm modulation with CRISPR-associated nucleases. In its present form, however, it has serious shortcomings that need be corrected before the manuscript is accepted for publication. The following major issues need to be addressed first.

Response: We thank the reviewer for their helpful input into our manuscript. Our responses are detailed after each of the points raised by the reviewer.

1. Statistics data suggests that multiplexed guides had higher killing efficiencies as a group than their single guide constituents, but there was an exception, rplC, with very low killing efficiency even use multiple guides. Is that because of the essentiality of this gene in *E. coli*? How about change the guide sequence identity with *E. coli*?

Response: Our revised manuscript includes data from 65 sgRNAs. The killing efficiencies ranged from 100% to ~1%. In the case of the rplC sgRNA, we re-sequenced the plasmid to find that the rplC sgRNA was duplicated. We subsequently re-cloned the rplC sgRNA and confirmed by sequencing that only a single sgRNA was inserted. The new rplC sgRNA (rplC.1) has lower killing efficiency than the original, duplicated sgRNA. We do not know why the rplC sgRNA (and others) have low killing efficiencies. We calculated a mis-match score for each sgRNA to the *E. coli* genome by searching for putative off-target sites with < 6 mis-matches to each sgRNA (Supplemental Table S3, Supplemental Files S4-S6). As discussed below, and in the responses to reviewer 2, killing efficiency is not correlated with identity to the *E. coli* genome.

2. The experiments study the influences of sgRNA identity to the conjugative donor genome on *S. enterica* killing efficiency are very preliminary. The authors gave very few examples on this experiment. Comprehensive design a pool of sgRNAs with different identity to *E. coli*, and test their influences on killing efficiency would be helpful.

Response: We have now tested a total of 65 sgRNAs for their killing efficiency in *S. enterica* that have a range of identities and targeting parameters. See also our response to point 3 below, and our response to the L140 concern raised by reviewer 2.

3. There have been reported sgRNA predictions model for prokaryotes (ref. 32 as cited by the authors), will that be helpful for the correlation of killing efficiency with predicted sgRNA activity? A predictable killing efficiency is always welcomed.

Response: Our original manuscript used a Church lab prediction algorithm that was based on *in vitro* data, and on Cas9 activity in mammalian cells. A new prediction algorithm (from a different group) optimized for prokaryotic genomes was recently published (Guo et al. 'Improved sgRNA design in bacteria via genome-wide activity profiling', NAR, <https://doi.org/10.1093/nar/gky572>). We used the Guo et al. algorithm in our revised manuscript. We thought this new algorithm was appropriate because the read-out of sgRNA activity was cell death - very similar to the killing efficiency measurements in our study. In short, we find a weak correlation between *S. enterica* killing efficiency and any single sgRNA parameter (predicted activity, position of sgRNA, strand being targeted, etc.). We used the different targeting parameters of our 65 sgRNAs in a generalized linear model to examine what parameters influenced sgRNA killing efficiency in *S. enterica* (Supplemental Figure S6). Two parameters were weakly correlated with killing efficiency - a positive correlation with predicted sgRNA activity and a negative correlation with targeting essential genes. A cautionary note. When using the Guo et al. algorithm to predict activity of our 65 sgRNAs, a random sgRNA sequence was equally likely to have high predicted activity as a 'real' sgRNA targeting a sequence in *S. enterica*. One major difference between the Guo et al. methodology and ours is the method of sgRNA/CRISPR delivery; plasmid transformation versus conjugative delivery. It is clear from our data that parameters that govern sgRNA targeting/activity in bacterial genomes are not completely understood, especially within the context of conjugative delivery. We agree that a comprehensive study of pools of sgRNA delivered by conjugative transfer may help us better understand targeting parameters. Such studies are now underway.

4. Minor point: Fig.1C was not cited in the paper. And I do not see it is informative in the paper.

Response: We have now referenced Figure 1C.

Reviewer 2 (Remarks to the Author):

The work reported in this manuscript describes a successful approach to improve the spread of the established CRISPR/Cas9 system, that serves as an anti-microbial agent, in a two-bacteria setup. The placement of the conjugation machinery and the CRISPR system on a conjugative plasmid leads to improved conjugation frequencies hence the improved spread of CRISPR/Cas9 system. The study is worth to publish, however, there are multiple shortcomings. Hence I conclude as 'major revision'.

Response: We thank the reviewer for their positive input about our paper.

Below I list both my major and minor comments:

Major comments:

1. I do not see any compelling evidence to justify the use of biofilm statement in the title. To me this study simply reports how the constructed pNuc-cis improves the spread of the CRISPR nuclease, via conjugation, as compared to the pNuc-trans. The use of CRISPR/Cas9 as a DNA-sequence-specific antimicrobial agent has already been reported hence the authors do not need to re-claim this point. The authors also do not provide : . any background to the biofilm aspect, except mentioning a condition 'mimicking biofilm' at the end of introduction. I encourage the authors to provide all the necessary background in the introduction text instead of taking too many shortcuts by citing other publications;

. any empirical finding to support that the use of beads leading to increased biofilm formation, except assuming that the increased surface will lead to biofilm formation. I can follow the logic, however, to have such a claim, use of biofilm in the title, experimental biofilm formation has to be provided.

Response: The use of solid surface and glass silica beads to promote biofilm formation has an extensive publication and citation history. In our original manuscript, we provided 5 references to experimental studies showing that addition of solid glass beads to bacterial cultures lead to the growth of bacteria in cell-to-cell contact, meeting the definition of a biofilm - dense microbial communities that grow on living or inert surfaces. In fact, silica beads of the type we used here were specifically shown to promote biofilm formation. Given the weight of the published literature on this subject, we did not feel it necessary to empirically demonstrate biofilm formation. We have removed specific references to biofilm from the revised manuscript, except for the discussion where it is appropriate to discuss biofilms in the context of the human microbiome (where most microbial communities are biofilms).

2. The study uses TevSp/Sa/Cas9 system however they do not provide any detail on the mode of action of the killing. The authors should provide all the necessary background to help the readers.

Response: We have added material to the introduction describing the mode of action of Cas9 killing (introduction of double-strand breaks).

3. The authors use inconsistent terminology throughout the manuscript that leads to confusion. For instance, due to the inconsistent terminology use, it is not clear to me what the authors aim to do with the developed system: Is the aim controlling, killing, modifying, modulating bacteria? All these can be achieved, however, this being a research article, and not a commentary, a better focus is necessary.

Response: In our view, these terms are interchangeable and all point to the potential

of conjugation delivery of molecular tools to bacteria. However, we recognize that readers may be confused by changes in terminology and thus have used 'modulation' or 'modulate' throughout the manuscript.

Another terminology: conjugation efficiency vs. conjugation frequency. The authors use both terminology interchangeably without a clear distinction. The authors do not perform any genetic engineering to the tra clusters/genes hence there is no direct attempt to improve the efficiency of conjugation. Testing different agitation speed and NaCl concentrations may justify the use of term 'conjugation efficiency', however, due to inconsistent term use I cannot distinguish it. What has been reported is the improved conjugation frequency, or the number of successful conjugation events (given what the conjugation frequency calculation is based on).

Response: This is a good point. We have changed the paper to use the term conjugation frequency. While it is true we have not specifically engineered the tra genes, we did engineer the conjugative plasmids to include the components necessary for cis transfer (which were not previously linked on the same plasmid).

4. The methods section is incomplete hence not satisfactory. There is a lack of focus to the important details (see my comments below to specific examples.) I cannot figure out, neither based on the text in the main body nor in the method section, what has been done for the construction of pNuc-cis. From the Figure 1a, it seems that the authors cloned the CRISPR system and the oriT sequence to pTA-Mob, however, there is a discrepancy between the description in the main body text and the method section. Overall, the method section should clearly describe the specific design of the study and provide clear and concise description of the procedures that were performed.

5. Given the number of plasmids constructed, it would be helpful to provide the nucleotide sequences of the plasmids (or plasmid maps). I would encourage the authors to provide the sequences in supplementary, as gene bank files, or deposit them to open public repositories for easy access.

Response to points 4 and 5: We have modified the text to include a better description of assembly (a detailed protocol is available in the supplement). We have included plasmid maps (Supplemental Fig S2) and annotated GenBank files of pNuc-cis, pNuc-trans and pTA-Mob (Supplemental Files S1-S3).

Minor comments:

L12. Microbial ecosystems are essential for human health and proper development, and disturbances of the ecosystem correlate with a multitude of diseases (15). This is a confusing sentence: the citation one is about rhizosphere (I do not see the immediate relevance of rhizosphere on human development), and two to five are about gut microbiomes. Would not it be easier by simply referring to gut microbiome instead of microbial ecosystems?

Response: Our intent was to point to the general importance of microbial communities in human health. We have added additional references that point to the general importance of the human microbiome for health, not just for the gut microbiome. The majority of human microbiome research has been conducted on the gut simply due to availability and accessibility of samples.

L20-21 composition of the microbial ecosystem composition of gut microbiome

Response: Fixed as suggested.

Inconsistent terminology for the set goal: L22, targeted modification of microbiomes; L34, targeted modulation of microbiomes; L42, tool for manipulation of microbiomes.

Response: We have modified the manuscript to use the term modulation throughout.

L28-30 Highly efficient. Please re-phrase the sentence. It is not clear what is efficient; is it the killing-effect or the conjugation (which rather should be frequency)?

L34 Here, we optimized conjugative delivery of CRISPR nucleases for targeted modulation of microbiomes. I do not understand the concept of modulation of microbiomes. This study only shows the use of CRISPR system as an antimicrobial agent. Hence the only control is the killing effect that has been reported for a single strain. Please re-phrase.

Response: We apologize for not being clear. As proof-of-concept, we tested our system in a mixed culture of two bacteria. Other studies that looked at conjugative systems used *E. coli* to *E. coli* model systems. It is not unreasonable to think that conjugative plasmids would spread within complex bacterial communities. Selectively removing one species (or multiple) species from this community using CRISPR would in effect modulate the abundance (or composition) of bacteria relative to each other. These ideas have been expanded upon in the discussion.

L36 conjugation machinery φ - φ conjugative machinery. Please stick to one.

Response: Changed to conjugative where appropriate.

L36-37 the conjugation machinery and nuclease were separated. It is not clear what separated means? Are they located on separate plasmids, or a certain part is on the host genome, or ?

Response: We have added the phrase "...the conjugation machinery and nuclease were encoded on different DNA molecules".

L38 re-conjugation conjugation.

Response: We used re-conjugation in this context because we wanted to highlight an important component of our cis plasmids, namely that recipient become subsequent donors for conjugation (re-conjugation). In the revised version of the manuscript, we have added experimental data to figure 1 showing that *S. enterica* pNuc-cis transcon-

jugants can act as donors to naive recipients. This observation supports our model that is shown in Figure 1c. We have modified the text in the appropriate places to re-phrase sentences uses the term 're-conjugate'.

L40 highly efficient conjugative transfer (of ?) The object is missing. L41 mimics a biofilm. There is no background provided to the biofilm aspect in the introduction? L41 highlight the promise of please simplify.

Response to Figure 1: Please note that Figure 1 has changed extensively from the original version. It now includes data on plasmid copy number and stability, as well as conjugation frequency of pNuc-cis from *S. enteric* transconjugants to naive recipients. Some of the points raised below are no longer relevant, and we have addressed those that address the new figure.

Figure 1 legend, the first line I am not familiar with the concept of using statements as figure names. Regardless, the statement is not clear. Efficient in what sense? Would not it be better to give the figure a name that reflects both the content and the message it aims to convey?

Figure 1 legend a. Schematic of the a. Schematic view of the.

Figure 1 legend conjugation genes There are no genes that are called conjugation genes hence the use transfer functions, genes required for conjugation or alike would suit better.

Response. Changed as requested.

Figure 1 legend from the pTA-Mob plasmid and derived from the IncP RK2 conjugative system. It is not clear, what is derived from what.

Response. This has been changed.

Figure 1 legend the b section in the legend is not describing all the elements that are depicted in the b part of the figure.

Response. Changed.

Figure 1 legend the section c is erroneous. Firstly, the legend does not describe the figure. Secondly, why does the number of donor cells (green circles) under the use of pNuc-trans is increasing over time? In the -trans setup the pTA-Mob is not mobile, hence to me the number of donors should not increase over time hence this part of the figure is confusing.

Response. The number of donor cells under pNuc-trans increases simply because over time all the cells in the culture will grow. This has been added to the legend to clarify this point.

Figure 1 Conjugation efficiency Please see my comment below on the inconsistent term use.

L44 higher levels of conjugation This is not clear as there is no highness of lowness to conjugation. A suggestion would be: A cis conjugative plasmid leads to higher conjugation frequency than a plasmid with trans setup.

Response: Changed as requested.

L45 We engineered a conjugative plasmid, pNuc, based on the IncP RK2 conjugative plasmid to examine parameters that contributed to conjugation (Fig. 1A). Please re-phrase. It is not clear what has been engineered. Do the authors refer to pNuc-cis? pNuc-cis looks as if it is based on pTA-Mob, but I do not see which part of pNuc-trans originates from pTA-Mob.

Response: This has been clarified.

L45 IncP RK2 conjugative plasmid Please re-phrase. The citation refer to the pTA-Mob plasmid, which is not conjugative as it does not harbour an oriT.

Response: We deleted the word conjugative.

L46 The pNuc plasmid encoded Please re-phrase, the plasmid contains/carries/harbours the nuclease, does not encodes it.

L48-50 and a single guide RNA (sgRNA) cassette driven by a constitutive promoter derived from the tetracycline resistance gene (pTet) into which we cloned oligonucleotides corresponding to predicted target sites in the *S. enterica* genome. Too long sentence, please break it up. Also it reads as if the oligos were cloned into the tet gene.

Response: To clarify the nature of the pTA-Mob construct, we have modified the sentence from lines 48 to 51.

L51-54 This is a lengthy and confusing sentence. Unless I missed it, the method section does not provide the details, the authors simply cloned both the CRISPR system and oriT into the pTA-Mob. Hence, if this is correct, I do not understand why the conjugation machinery and the gentamicin resistance gene were cloned into the pTA-Mob?

Response: To clarify the nature of the pTA-Mob construct, we have modified the sentence from lines 48 to 51.

L52; L56 the conjugative assistance plasmid, pTA-Mob; pTA-Mob mobilization plasmid Please be consistent with the terminology use. I would opt for the term 'conjugative assistance/helper plasmid'.

Response: To clarify the nature of the pTA-Mob construct, we have modified the sentence from lines 48 to 51.

L61 conjugation efficiency This refers to the conjugation frequency and not efficiency. This applies to the entire text.

Response: Global changes made.

L64 This result suggests pNuc-trans is lost from a percentage of exconjugants after the initial burst of conjugation, possibly because it is toxic or unstable. Why do the authors claim that pNuc-trans was lost from the recipients? I cannot see any experimental findings showing the actual number of donors that are Cm resistant (carrying the pNuc-trans plasmid) over the course of the conjugation experiment. In the absence of such data this is not justifiable.

Response: We have now added data to Figure 1 on plasmid stability after conjugation, and stability in the *E. coli* donor strain, both in the absence of antibiotic selection. This data

L75 conjugation efficiency conjugation frequencies L76 conjugation efficiency conjugation frequency.

Response. Changed as requested.

Figure 2 legend circles circles

Response. Changed as requested.

Figure 2 I imagine the TevSpCas9 was used to generate the data, but there is nothing mentioned specifying which of the Cas9 system was used to generate the data used in figure 2.

Response: All of the data shown in Figure 2 is with the TevSpCas9 nuclease. This has been added to the legend.

L82 increase in conjugation increase in conjugation frequency

Response. Changed as requested.

L82 cell-cell contact cell-to-cell contact

Response. Changed as requested.

L83 The previous experiments demonstrated that pNuc-cis was more efficient (in compared to what) in a filter mating assay on solid media.

Response. Changed as requested.

L105 TevSa/SpCas9 There is no background provided to neither of the Cas9 systems. What are their differences and why have they been used? Please help the readers to follow your experimental logic.

Response: We have added text to the manuscript explaining the difference between the two Cas9 nucleases.

L110 A Mann-Whitney test showed The details of this statistics analysis has not been provided in the methods section.

Response: This section of the manuscript has changed. Details of the Mann-Whitney test are included in the supplemental methods.

L118 growth phenotype There is no detail provided on how the growth was measured, under what condition(s). There is also no corresponding section in the materials and methods part.

Response: This section of the manuscript has changed.

L139 the gene the coding sequence? Please be specific wherever possible as 'gene' is a rather vague term.

Response. Changed as requested.

L140 predicted sgRNA activity I do not understand the term 'activity' here. Different sgRNA simply target different positions within the coding sequence of the gene. There is no detail provided on the output of the PERL script hence I cannot understand what is meant by activity. Please specify.

Response: Please see also our response to point 3 of reviewer 1. This section of the manuscript has changed. The original predictions of sgRNA binding were done using a Church lab algorithm that was developed using *in vitro* data, and *in vivo* data from Cas9 activity in eukaryotic cells. Another sgRNA prediction algorithm was published that was optimized for sgRNA activity predictions in prokaryotic cells (Guo et al. NAR, 2018). In the paper describing the development of this prediction algorithm, the experimental readout was *E. coli* cell death. The authors use the term activity, but activity encompasses many sgRNA parameters - stability, expression, off-target binding, etc. We feel that our use of the term sgRNA activity is appropriate because our readout is killing efficiency, very similar to the readout from the Guo et al. study. Correlation between our experimental readout versus predicted activity is presented in Figure 4.

L157 IncP RK2 conjugative plasmid Please see my L45 comment above.

L160 key facet - we used key facet'em-dash'we used

Response. Changed as requested.

L175 broad-host range broad host range, please stick to one.

Response. Changed as requested.

L179, L187 pNUC pNuc

Response. Changed as requested.

L241 I assume that the *S. enterica* has KanR phenotype? No background information has been provided on this point. Please help the readers.

Response. Yes, *S. enterica* has a KanR phenotype. This information has been added

to the Methods section, and in the legend to Figure 1.

L242 exconjugant cells under TevSpCas9 repressive conditions (kanamycin 50 g/mL, chloramphenicol 25 g/mL, and 0.2% glucose) This part of the methodology only applies to pNuc-trans, since pNuc-cis does not harbour a CmR gene? If yes, then the details of the pNuc-cis is missing.

L243; 245 exconjugant cells expressing TevSpCas9; killing efficiency experiments I do not understand what the difference is? These are two different protocols for the same experiment? When the arabinose is present in the medium wont they become the same?

Response. We have substantially changed to Methods section to be more descriptive, including details of plasmid construction and antibiotics used under different conditions.

L270 plasmid expression for analysis and re-isolated for analysis. Please re-phrase.

Response. Modified to be clear.

Supplementary Figures. Update the title used in the document.

Also the file names are not human understandable. I would prefer file names such as Supplementary Figure 1 and so on. However I do not know who to blame, whether the submission system of the journal or the authors. Regardless, this makes the review even more challenging.

Response. This likely occurred during uploading. We have included a new single PDF of the Supplementary Information that includes the Supplementary Methods, Supplementary Figures and legends, a list of the Supplementary Tables, and a list of the Supplementary Files.

To conclude, the entire text needs a substantial editing prior to re-submission.

Rahmi Lale

REVIEWERS' COMMENTS:

Reviewer #1 (Remarks to the Author):

After a through reading of the new document, and the answers to my comments, this reviewer is convinced that the paper has improved considerably. The authors have shown a nice set of results, which conclusions are supported.

Chong Zhang

Reviewer #2 (Remarks to the Author):

Overall the comments by myself and the Reviewer 1 have been addressed appropriately, thank you. There are a few minor points remains, though, which I am listing below line-by-line. A final note, I might have not spotted all but I could still see the use of phrase 'conjugation efficiency' instead of 'conjugation frequency' in parts of the text. Please ensure the right use of the phrase throughout the main as well as the supplementary text, figures and legends.

Minor points:

L33, broad host -> broad-host

L80, Supplementary Figure S2 -> Supplementary Figure S2B (You may also consider swapping part A and B, following the order of cross-referencing in the main text).

L80, the main body text use the phrase conjugation frequency whereas the supplementary figure legend has the phrase conjugation efficiency. Please update.

L81-82, Supplementary Figure S2 -> Supplementary Figure S2A

L84, Supplementary Figure S2 -> Supplementary Figure S2C

L85, more efficient at conjugation -> The conjugation machinery is same in both systems, hence the better performance of the pNuc-cis is not due to being efficient in conjugation. As explained in the following text, it is due to occurrence of re-conjugation events therefore the text should rather read higher conjugation frequency. Please re-phrase.

L91, pNuc-cis was more efficient at conjugation -> Same as above comment to L85, please re-phrase.

L98, conjugation efficiency -> conjugation frequency

L116, 93 ± 8 -> missing %

L127, template or non-strand strands -> difficult to read, I suggest 'sense or antisense strands'

L156, key facet - we -> key facet'em dash (—) not a hyphen'we

L157, efficient conjugation -> increased occurrence of conjugation events

L159, conjugation efficiency -> conjugation frequency

L162, conjugation efficiencies -> conjugation frequencies

L194, (Δ hflA::Kan^R) -> Kan^R

L206, same plasmid -> which?

L211, transformed -> how, by which method?

L214, 5' -> substitute apostrophe with a prime sign (applies to the entire text)

Rahmi Lale

REVIEWERS' COMMENTS:

Reviewer 1 (Remarks to the Author):

After a through reading of the new document, and the answers to my comments, this reviewer is convinced that the paper has improved considerably. The authors have shown a nice set of results, which conclusions are supported.

Chong Zhang

Response: We thank the reviewer for the time and effort, and the positive comments about our manuscript.

Reviewer 2 (Remarks to the Author):

Overall the comments by myself and the Reviewer 1 have been addressed appropriately, thank you.

Response: We thank the reviewer for the time and effort, and the positive comments about our manuscript.

There are a few minor points remains, though, which I am listing below line-by-line. A final note, I might have not spotted all but I could still see the use of phrase 'conjugation efficiency' instead of 'conjugation frequency' in parts of the text. Please ensure the right use of the phrase throughout the main as well as the supplementary text, figures and legends.

Response: A global replacement of 'conjugation efficiency' for 'conjugation frequency' was done.

Minor points: L33, broad host → broad-host

Response: Changed as suggested.

L80, Supplementary Figure S2 → Supplementary Figure S2B (You may also consider swapping part A and B, following the order of cross-referencing in the main text).

Response: Changed as suggested. We followed the Nature guidelines and removed the 'S' for this and all following Supplementary Figure references.

L80, the main body text use the phrase conjugation frequency whereas the supplementary figure legend has the phrase conjugation efficiency. Please update.

Response: Changed as suggested.

L81-82, Supplementary Figure S2 → Supplementary Figure S2A

Response: Changed as suggested.

L84, Supplementary Figure S2 → Supplementary Figure S2C

Response: Changed as suggested.

L85, more efficient at conjugation → The conjugation machinery is same in both systems, hence the better performance of the pNuc-cis is not due to being efficient in conjugation. As explained in the following text, it is due to occurrence of re-conjugation events therefore the text should rather read higher conjugation frequency. Please re-phrase.

Response: Changed as suggested.

L91, pNuc-cis was more efficient at conjugation → Same as above comment to L85, please re-phrase.

Response: Changed as suggested.

L98, conjugation efficiency → conjugation frequency

Response: Changed as suggested.

L116, 938 missing %

Response: Added the % sign.

L127, template or non-strand strands → difficult to read, I suggest 'sense or antisense strands'

Response: Changed as suggested.

L156, key facet - we → key facet'em dash () not a hyphen'we

Response: Changed as suggested.

L157, efficient conjugation → increased occurrence of conjugation events

Response: Changed as suggested.

L159, conjugation efficiency → conjugation frequency

Response: Changed as suggested.

L162, conjugation efficiencies → conjugation frequencies

Response: Changed as suggested.

L194, (hilA::KanR) → Kan^R

Response: Changed as suggested.

L206, same plasmid → which?

Response: Changed to reference the plasmid.

L211, transformed → how, by which method?

Response: Electroporation. This was added to text.

L214, 5 → substitute apostrophe with a prime sign (applies to the entire text)

Response: Changed globally.